

# TravelBuddy
## Mobilny planer turystyczny

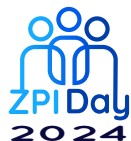

**Autorzy**: Igor Banaszak⊙ · Adrian Broniecki⊙ · Bartosz Gableta⊙ · Mateusz Gazda⊙
**Opiekun:** Marcin Kawalerowicz

**Streszczenie**

*TravelBuddy* to aplikacja mobilna zaprojektowana w celu uproszczenia procesu planowania podróży i zwiedzania atrakcji turystycznych. Głównym celem projektu było stworzenie narzędzia dostosowanego do indywidualnych potrzeb użytkowników, uwzględniającego przy tym takie aspekty jak dostępność dla osób z niepełnosprawnościami, preferencje dietetyczne czy potrzeby podróżujących ze zwierzętami. Aplikacja umożliwia zarządzanie planami wycieczek oraz oferuje rekomendacje atrakcji. Zintegrowano ją z narzędziami, takimi jak mapy i kalendarz, co ułatwia organizację podróży. Dodatkowe funkcje, takie jak tryb wysokiego kontrastu i tryb nocny, poprawiają dostępność i komfort użytkowania. *TravelBuddy* ma na celu poprawę komfortu planowania podróży, czyniąc ją bardziej dostępną dla różnorodnych grup społecznych.

## 1 WSTĘP

### 1.1 Opis problemu

Dynamiczny rozwój aplikacji mobilnych do planowania podróży oraz rosnące oczekiwania użytkowników w zakresie personalizacji i wygody wskazały na istotną lukę na rynku: brak kompleksowego narzędzia, które mogłoby efektywnie wspierać organizację podróży, uwzględniając indywidualne potrzeby użytkowników. Dostępne rozwiązania często nie spełniają oczekiwań w zakresie optymalizacji planów wycieczek, integracji z innymi narzędziami czy dokładnego oszacowania kosztów. Nasz zespół postanowił wykorzystać tę okazję, by zaprojektować rozwiązanie, które mogłoby skutecznie sprostać tym wyzwaniom.

### 1.2 Cele projektu

Głównym celem projektu *TravelBuddy* było opracowanie koncepcji aplikacji mobilnej, która będzie kompleksowo wspierać użytkowników w organizacji podróży. Projekt zakładał, że aplikacja będzie oferowała takie funkcje jak: zarządzanie planami wycieczek, dodawanie punktów wycieczki (ręcznie, na podstawie wyszukiwania atrakcji i na podstawie rekomendacji), uwzględnianie indywidualnych preferencji użytkowników przy rekomendacjach, a także integracja z kalendarzem i mapami.

Zespół miał na celu zbudowanie rozwiązania, które mogłoby odpowiadać na rosnące wymagania rynku turystycznego, oferując użytkownikom większą wygodę i personalizację. Technicznym celem projektu było także zaprojektowanie architektury aplikacji opartej na nowoczesnych technologiach, które mogłyby zapewnić skalowalność, wydajność i bezpieczeństwo.

Celem zespołu było, aby realizacja projektu *TravelBuddy* przyniosła istotne korzyści zarówno dla użytkowników, jak i dla biznesu. Uproszczenie procesu planowania podróży oraz personalizacja oferowanych rozwiązań zwiększy satysfakcję użytkowników i ich zaangażowanie w korzystanie z aplikacji. Jednocześnie projekt mógłby stworzyć możliwość współpracy z partnerami biznesowymi, takimi jak biura podróży czy dostawcy usług turystycznych, oferując im narzędzie do lepszego docierania do klientów i promowania swoich usług.

## 2 PRACE ZWIĄZANE Z TEMATEM

### 2.1 Istniejące rozwiązania i technologie

Na rynku istnieje wiele aplikacji wspomagających planowanie podróży, takich jak *TripIt* czy *TripAdvisor*, które oferują funkcje planowania wycieczek, wyszukiwania atrakcji i integracji z mapami. Jednak żadne z tych rozwiązań nie dostarczają dobrych rozwiązań do personalizacji w zakresie planowania wycieczek, szczególnie jeśli chodzi o uwzględnianie indywidualnych preferencji i wymagań użytkowników. Aplikacja

*TravelBuddy* powstała, aby uzupełnić tę lukę, oferując kompleksowe narzędzie do planowania podróży. Umożliwia ona zarządzanie planami wycieczek, rekomendowanie atrakcji w oparciu o określone kryteria oraz uwzględnianie specyficznych wymagań użytkowników.

## 2.2   Technologie

Aplikacja została zaprojektowana z myślą o urządzeniach z systemem Android (od wersji 6.0 wzwyż), co umożliwia dotarcie do szerokiego grona użytkowników i zapewnia kompatybilność z popularnymi urządzeniami mobilnymi. Rozwiązania technologiczne dobrano tak, aby zapewnić wysoką wydajność, elastyczność oraz bezpieczeństwo w użytkowaniu.

- Backend: Backend aplikacji oparto na ASP.NET Core [1], co gwarantuje wysoką wydajność, skalowalność oraz łatwość integracji z innymi usługami. Dzięki tej technologii możliwe było implementowanie zaawansowanych funkcji związanych z zarządzaniem danymi i logiką aplikacji.

- Frontend: Do realizacji warstwy frontendowej wybrano React Native [2] w środowisku Expo [3], co pozwala na tworzenie natywnych interfejsów użytkownika przy jednoczesnym wykorzystaniu jednego kodu źródłowego dla wielu platform mobilnych. Dzięki temu w przyszłości potencjalne rozszerzenie aplikacji na inne platformy będzie łatwiejsze.

- Infrastruktura chmurowa: Pierwotnie do zarządzania zasobami oraz obsługi autoryzacji użytkowników wykorzystano infrastrukturę Microsoft Azure [4], w tym usługę Azure AD B2C. Jednakże napotkane ograniczenia w zakresie elastyczności i integracji tej platformy skłoniły zespół do migracji na rozwiązanie oferowane przez Amazon Web Services – AWS Cognito [5]. AWS Cognito okazało się bardziej dopasowane do specyficznych potrzeb aplikacji, oferując większe możliwości konfiguracji, co umożliwiło m.in. zaimplementowanie formularzy wbudowanych w aplikację. Oprócz usług AWS do projekt wykorzystuje także usługi Azure, takie jak Azure Container Instances do wdrażania kontenerów dockerowych [6], czy Azure SQL Database Server.

- Integracja z zewnętrznymi źródłami danych: Aplikacja korzysta z API Geoapify [7], które stanowi źródło danych dotyczących atrakcji turystycznych. Dzięki temu użytkownicy mają dostęp do szerokiej bazy informacji wspierających planowanie podróży. Zespół zadbał o filtrowanie i weryfikację pozyskiwanych danych, aby zapewnić ich aktualność oraz wysoką jakość. Wykorzystujemy także API Narodowego Banku Polskiego, aby umożliwić użytkownikom możliwość śledzenia wydatków związanych z wycieczką w wybranej przez nich walucie.

Takie podejście umożliwiło stworzenie aplikacji, która łączy nowoczesne technologie z praktycznymi funkcjonalnościami, spełniając potrzeby zarówno użytkowników końcowych, jak i wymogi techniczne niezbędne do dalszego rozwoju produktu.

## 2.3   Metodyka wytwarzania oprogramowania

W realizacji projektu zastosowano zwinne podejście do wytwarzania oprogramowania, oparte na metodyce Scrum. Prace zostały podzielone na cztery sprinty: trzy sprinty dwutygodniowe i jeden sprint trzytygodniowy. Do zarządzania zadaniami i monitorowania postępów wykorzystano narzędzie Jira. Takie podejście zapewniło elastyczność w realizacji projektu oraz efektywne wykorzystanie dostępnego czasu i zasobów.

## 2.4   Czas

Projekt był realizowany w ramach ograniczonego harmonogramu, co wymusiło priorytetyzację kluczowych funkcjonalności:

- Zarządzanie planami podróży.

- Rekomendacje atrakcji.

- Dodatkowe funkcje ułatwiające organizację podróży, takie jak: powiadomienia, integracja z kalendarzem i mapami.

- Opcje dostosowania aplikacji do użytkownika: tryb nocny i tryb wysokiego kontrastu.

## 2.5  Ograniczenia technologiczne

W trakcie realizacji projektu napotkano następujące ograniczenia technologiczne:

· Ograniczony budżet - realizując projekt zespół chciał zminimalizować koszty, co ograniczyło wybór narzędzi, usług i infrastruktury. Zdecydowano się na wykorzystanie darmowych planów, planów studenckich oraz rozwiązań open-source, co w niektórych przypadkach wiązało się z ograniczeniami w funkcjonalności i skalowalności.

· Ograniczenia środowiska Expo - Expo, jako framework do budowy aplikacji natywnych w JavaScript, znacznie ułatwia proces deweloperski, ale ogranicza możliwość implementacji niektórych funkcji natywnych (np. zaawansowanych modyfikacji UI specyficznych dla systemów iOS lub Android). W przypadku aplikacji wymagających dostosowania bibliotek do funkcji specyficznych dla platform mobilnych, brak wsparcia Expo dla niektórych bibliotek natywnych wymagał poszukiwania alternatywnych rozwiązań lub rezygnacji z funkcji.

· Testowanie i debugowanie aplikacji - wykorzystywanie emulatorów zamiast rzeczywistych urządzeń z powodu ograniczeń sprzętowych mogło powodować różnice w działaniu aplikacji w środowisku testowym i produkcyjnym. Miało to również wpływ na szybkość wdrażania funkcjonalności.

· Kompletność danych o atrakcjach – dane dostarczane przez Geoapify nie zawsze są kompletne, zwłaszcza w mniej zurbanizowanych obszarach. Może to negatywnie wpływać na jakość planowania wycieczek w takich regionach, ponieważ brak pełnych informacji o dostępnych atrakcjach może prowadzić do pominięcia ciekawych miejsc lub wyboru punktów, które nie spełniają oczekiwań użytkownika.

# 3  WYNIKI

## 3.1  Zrealizowane funkcjonalności

Zaimplementowano następujące funkcjonalności:

· Zarządzanie kontem – tworzenie konta, logowanie, resetowanie hasła

· Tworzenie i zarządzanie wycieczkami – planowanie wycieczek z możliwością dodawania, edytowania, przeglądania oraz usuwania wycieczek

· Preferencje i udogodnienia – tworzenie i zarządzanie profilami preferencji oraz udogodnień, które wpływają na rekomendacje otrzymywane w trakcie planowania

· Zarządzanie punktami wycieczki – dodawanie punktów manualnie, na podstawie wyszukiwania lub rekomendacji, a także edycja szczegółów punktów

· Śledzenie i monitorowanie budżetu – zarządzanie wydatkami związanymi z punktami oraz całymi wycieczkami

· Integracja z zewnętrznymi narzędziami – dodawanie punktów wycieczki do kalendarza oraz nawigowanie pomiędzy punktami trasy

· Przypomnienia – ustawianie przypomnień o nadchodzących punktach wycieczek

## 3.2 Interfejs użytkownika

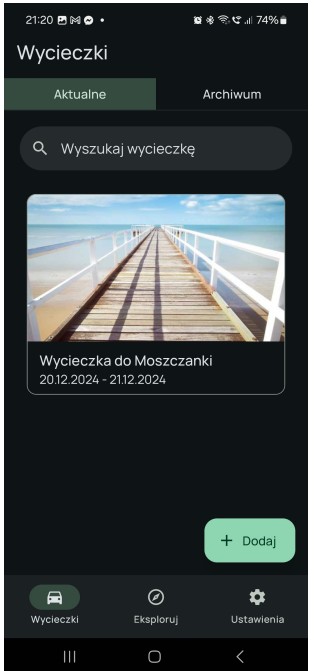

(a) Przegląd wycieczek

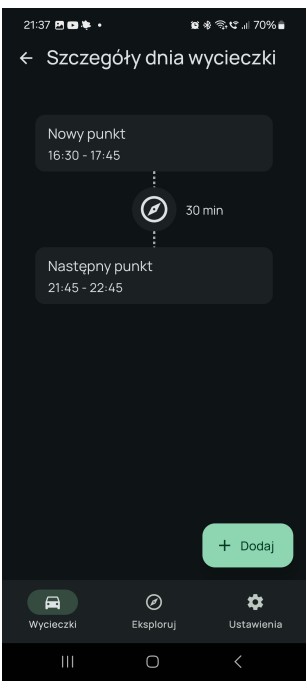

(b) Widok dnia wycieczki

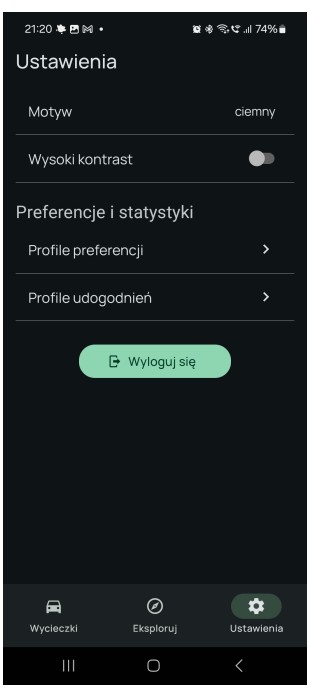

(c) Ustawienia

## 3.3 Analiza osiągniętych celów biznesowych i technologicznych

Projekt *TravelBuddy* zrealizował swoje główne założenia, tworząc aplikację wspierającą użytkowników w kompleksowym planowaniu wycieczek. Opracowane rozwiązanie uwzględnia kluczowe funkcjonalności, takie jak zarządzanie planami podróży, dodawanie punktów wycieczek, integracja z kalendarzem i mapami oraz personalizacja na podstawie preferencji użytkownika. Dzięki temu aplikacja spełnia podstawowe potrzeby użytkowników w zakresie organizacji podróży.

Jednym z osiągnięć projektu było zaimplementowanie systemu rekomendacji atrakcji turystycznych. System ten, bazując na analizie miejsca docelowego, preferencji użytkownika i wymaganych udogodnień, umożliwia proponowanie miejsc zgodnych z oczekiwaniami użytkownika. Niemniej jednak, ze względu na ograniczony zakres analizy, funkcjonalność ta pozostaje stosunkowo podstawowa. Aby poprawić skuteczność i precyzję rekomendacji, w przyszłości należałoby rozważyć rozszerzenie systemu o dodatkowe dane, takie jak historia odwiedzonych miejsc, opinie innych użytkowników czy dynamiczne czynniki, takie jak aktualna pogoda czy lokalne wydarzenia.

Aplikacja została zaprojektowana z uwzględnieniem nowoczesnych standardów technologicznych, co zapewnia skalowalność, wydajność i bezpieczeństwo. Otwiera to możliwość dalszego rozwoju, w tym integracji z usługami partnerów biznesowych, takich jak biura podróży czy dostawcy usług turystycznych.

## 3.4 Ocena możliwości wdrożenia projektu w kontekście przemysłowym

Działanie aplikacji zostało przetestowane w środowisku zbliżonym do produkcyjnego. Aplikacja była instalowana na fizycznych urządzeniach mobilnych za pomocą pliku `.apk` i komunikowała się z serwerem uruchomionym w chmurze. Taka konfiguracja umożliwiła weryfikację podstawowych założeń funkcjonalnych i technicznych projektu, jak również identyfikację obszarów wymagających dalszych ulepszeń.

Jednym z istotnych elementów, które należałoby rozwinąć przed wdrożeniem produkcyjnym, jest system dystrybucji aplikacji. Obecny proces instalacji ręcznej za pomocą plików `.apk` jest odpowiedni na etapie testów, jednak w kontekście przemysłowym konieczne byłoby przygotowanie aplikacji do publikacji w oficjalnym sklepie, takim jak Google Play. Dzięki temu aplikacja będzie łatwiej dostępna dla użytkowników końcowych, a proces jej aktualizacji i utrzymania stanie się bardziej efektywny.

## 4   WNIOSKI

Wyniki osiągnięte w projekcie *TravelBuddy* potwierdzają, że aplikacja spełnia kluczowe cele, oferując funkcjonalności, które wspierają użytkowników w planowaniu wycieczek, takie jak zarządzanie wycieczkami, personalizowane rekomendacje atrakcji oraz integracja z narzędziami zewnętrznymi. Aplikacja oferuje wygodę i inkluzywność, wspierając szerokie grono odbiorców, w tym osoby z różnymi potrzebami. Dzięki temu jest atrakcyjna zarówno dla użytkowników indywidualnych, jak i potencjalnych partnerów biznesowych, takich jak biura podróży czy dostawcy usług turystycznych. Chociaż aplikacja spełnia główne założenia, nadal istnieje przestrzeń do udoskonalenia niektórych funkcji, szczególnie w zakresie zaawansowanych rekomendacji i integracji z dodatkowymi źródłami danych. Mimo tych wyzwań projekt stanowi solidną podstawę do dalszego rozwoju i wdrożenia na szerszą skalę.

## 5   KIERUNKI ROZWOJU

Możliwe kierunki rozwoju *TravelBuddy* obejmują:

- Integrację z dodatkowymi źródłami danych - zamiast polegać na jednym źródle (Geoapify), aplikacja mogłaby integrować się z innymi źródłami danych o atrakcjach turystycznych

- Wykorzystanie sztucznej inteligencji - aplikacja mogłaby zostać rozszerzona o funkcje oparte na sztucznej inteligencji, m.in. do wykrywania i korekcji nieprawidłowych danych, rekomendowania atrakcji, czy też do tworzenia całych planów wycieczek

- Stworzenie bardziej zaawansowanych algorytmów rekomendacji - aplikacja mogłaby analizować historię użytkownika w celu dostarczenia jeszcze bardziej trafnych rekomendacji

- Integrację z mediami społecznościowymi - umożliwienie użytkownikom dzielenia się swoimi planami podróży i doświadczeniami mogłaby zwiększyć popularność aplikacji

- Rozbudowę o funkcjonalności offline - funkcje planowania i nawigacji bez dostępu do internetu umożliwiłyby korzystanie z aplikacji w rejonach, gdzie dostęp do sieci jest ograniczony

## 6   PODZIĘKOWANIA

Pragniemy złożyć wyrazy najgłębszej wdzięczności dla naszego opiekuna, doktora Marcina Kawalerowicza, za nieocenione wsparcie i zaangażowanie podczas tworzenia projektu. Serdecznie dziękujemy także opiekunom pozostałych grup, doktor Bernadetcie Maleszce oraz doktorowi Marcinowi Pietranikowi, z którymi mogliśmy konsultować nasze pomysły i rozwiązania. Podziękowania kierujemy również do profesora Piotra Bródki, za cenne wskazówki podczas zajęć seminaryjnych.

## LITERATURA

[1] *ASP.NET Core Documentation*, Dostęp: 11.2024. `https://learn.microsoft.com/en-us/aspnet/core/?view=aspnetcore-9.0&WT.mc_id=dotnet-35129-website`.

[2] *React Native Documentation*, Dostęp: 11.2024. `https://reactnative.dev/docs/getting-started`.

[3] *Expo Documentation*, Dostęp: 11.2024. `https://docs.expo.dev/`.

[4] *Azure Documentation*, Dostęp: 11.2024. `https://learn.microsoft.com/pl-pl/azure/?product=popular`.

[5] *AWS Cognito Documentation*, Dostęp: 11.2024. `https://docs.aws.amazon.com/cognito/`.

[6] *Docker Documentation*, Dostęp: 11.2024. `https://docs.docker.com/`.

[7] *Geoapify*, Dostęp: 11.2024. `https://www.geoapify.com/`.