# OpenReview forum: "TravelBuddy - mobilny planer turystyczny"
_pwr.edu.pl/Wrocław_University_of_Science_and_Technology/2024/ZPI_Day — Wrocław University of Science and Technology 2024 ZPI Day Submission_

### Official Review · Reviewer_pi3q · 2024-12-05
**Rekomendowany do publikacji po drobnych poprawkach**

**Confidence:** 5
**Significance Of Results:** 4
**Overall Quality:** 4

**Compliance With Template:**

5: Very High Quality – The article contains all the required sections, which are written in a very detailed, clear, and error-free manner. The structure is professional and meets expectations, and the content adheres to the highest substantive and formal standards.

**Description Of Results:**

4: High Quality – The results are described in detail and supported by usage examples or evaluations. The description is reliable but may lack full depth of analysis.

**Feedback On Consistency:**

Tekst jest spójny i logiczny, choć niektóre przejścia między sekcjami mogłyby być płynniejsze:
1. np. przejście z 2.4 do 2.5 do 3 to przejście z listy do listy; warto uzupełnić tekst i zadbać o lepszą narrację (np. połączyć 2.4 z 2.5 i uzupełnić, o akapit płynnie łączący ograniczenia czasowe z technologicznymi)
2. 3.1. - uzupełnić kropki na końcu elementów listy
3. 3.2 - uzupełnić o krótką informację tekstową co widzimy (to nie musi być osobna sekcja, to może być ilustracja do 3.1.)
4. 3.2. - środkowy zrzut wydaje się pokazywać testowe dane; warto wymienić go na coś z bardziej pokazującego konkretny przykład (na zrzucie jest sporo czarnego tła niewnoszącego niczego ciekawego do wiedzy o projekcie)

**Potential For Development:**

Artykuł dobrze opisuje kierunki rozwoju oraz możliwości wdrożenia praktycznego.

**Project Nature Evaluation:**

Opisany projekt ma dobrze opisaną naturę pracy inżynierskiej. Jest użyteczny (odpowiada na konkretne potrzeby). Wykorzystuje spójny stos technologiczny. Podczas pracy użyto nowoczesnych rozwiązań: agile, scrum, testy. W projekcie zintegrowano istniejące API (Geoapify, NBP).

Być może warto byłoby dodać do artykułu diagram przedstawiający architekturę rozwiązania uwzględniającą wewnętrzną budowę systemu oraz integracje z zewnętrznymi API.

**Technical Language Precision:**

4: High Quality – The language is appropriate for a technical report. Terminology is used correctly, and statements are precise, with only minor shortcomings that do not affect the overall clarity.

---

### Official Review · Reviewer_cmP4 · 2024-12-05
**TravelBuddy - Mobilny planer turystyczny**

**Confidence:** 5
**Significance Of Results:** 5
**Overall Quality:** 4

**Compliance With Template:**

5: Very High Quality – The article contains all the required sections, which are written in a very detailed, clear, and error-free manner. The structure is professional and meets expectations, and the content adheres to the highest substantive and formal standards.

**Description Of Results:**

4: High Quality – The results are described in detail and supported by usage examples or evaluations. The description is reliable but may lack full depth of analysis.

**Feedback On Consistency:**

The paper is consitent, written very clearly with good description of many technical aspects of both the product and the project. Motivation of the technical decisions made by the team is also explained very well. However, information of the approach to the product quality verification is almost not present.

**Potential For Development:**

Both practical applications of the system and the possibilites for further work are described in the paper very widely. The authors identifed the gap on the market related to trip planning and they addressed just this map. As the result they created a product which is already competitive on the market and they aslo indicated and planned the evolution of the product. They are planning both to exted functionalities, interoperability with the external systems and to improve the quality of the algorithms implemented in the system.

**Project Nature Evaluation:**

The product has high potential for commercial usage especially after adding planned functionalities. It is scalable, well designed, the product reuses solutions available on the market to handle some functional areas of the project system implemented by the authors. The product is based and comunicates to the external solutions which are novel. These decisions helped to save efforts of the project team and to focus on the most important aspects of the project, which are not available on the market.

**Technical Language Precision:**

5: Very High Quality – The language is entirely appropriate for a technical report. All terms are used correctly and precisely, and the style is professional, clear, and coherent, without any errors or ambiguities.

---

### Official Review · Reviewer_umiW · 2024-12-05
**Mobilny planer turystyczny**

**Confidence:** 5
**Significance Of Results:** 3
**Overall Quality:** 3

**Compliance With Template:**

3: Average Quality – The article includes most of the required sections, but some may be incomplete, written in a general or unclear manner. The content is correct but requires further refinement.

**Description Of Results:**

3: Average Quality – The results are described with moderate detail. Some examples or evaluation elements are present but insufficiently developed or incomplete.

**Feedback On Consistency:**

Wyniki powinny być bardziej rozwinięte. Autorzy wyniki wymienili tylko w punktach.

**Potential For Development:**

Aplikacja ma potencjał wdrożenia komercyjnego.

**Project Nature Evaluation:**

Projekt wykazuje cechy pracy inżynierskiej. Dla konkretnej potrzeby biznesowej zastosowano właściwe narzędzia informatyczne.

**Technical Language Precision:**

4: High Quality – The language is appropriate for a technical report. Terminology is used correctly, and statements are precise, with only minor shortcomings that do not affect the overall clarity.

---

### Official Review · Reviewer_X2Gy · 2024-12-06
**Aplikacja stanowi ciekawe rozwiązanie w kontekście wsparcia w planowaniu wycieczek, posiada trafne rozwiązania pod względem użytkowym oraz uwzględnia różnorodność preferencji.**

**Confidence:** 3
**Significance Of Results:** 5
**Overall Quality:** 4

**Compliance With Template:**

5: Very High Quality – The article contains all the required sections, which are written in a very detailed, clear, and error-free manner. The structure is professional and meets expectations, and the content adheres to the highest substantive and formal standards.

**Description Of Results:**

4: High Quality – The results are described in detail and supported by usage examples or evaluations. The description is reliable but may lack full depth of analysis.

**Feedback On Consistency:**

Zaprezentowany projekt identyfikuje lukę, którą przedstawione rozwiązanie może w pewnym stopniu zaspokoić. Warto docenić analizę możliwych do zastosowania rozwiązań technicznych jak i ograniczeń z tym związanych. W kontekście przedstawionych wyników zastosowano ogólne określenia np. zarządzanie profilami preferencji, jednak nie wiadomo czy aplikacja uwzględnia różne rodzaje transportu.

**Potential For Development:**

W projekcie przedstawiono możliwość integracji z zewnętrznymi narzędziami np. kalendarzem i dodawanie punktów wycieczki do kalendarza. Można w tym obszarze poszukać możliwości, by aplikacja uwzględniała zajętość kalendarza w kontekście planowania np. wycieczka musi się skończyć w takim terminie, bo użytkownik ma zaplanowany wyjazd służbowy itd.

**Project Nature Evaluation:**

Zastosowane w projekcie rozwiązania technologiczne oraz wysoki poziom użyteczności kwalifikuje przedstawioną pracę jako spełniającą cechy pracy inżynierskiej.

**Technical Language Precision:**

4: High Quality – The language is appropriate for a technical report. Terminology is used correctly, and statements are precise, with only minor shortcomings that do not affect the overall clarity.

---

### Decision · Program_Chairs · 2024-12-10

Accept (Poster)